# The Promise of Patient-Derived Preclinical Models to Accelerate the Implementation of Personalised Medicine for Children with Neuroblastoma

**DOI:** 10.3390/jpm11040248

**Published:** 2021-03-30

**Authors:** Elizabeth R. Tucker, Sally George, Paola Angelini, Alejandra Bruna, Louis Chesler

**Affiliations:** 1Paediatric Tumour Biology, Division of Clinical Studies, The Institute of Cancer Research, Cotswold Road, London SM2 5NG, UK; lizzie.tucker@icr.ac.uk (E.R.T.); sally.george@icr.ac.uk (S.G.); 2Children and Young People’s Unit, The Royal Marsden, Downs Road, Sutton, Surrey SM2 5PT, UK; paola.angelini@nhs.net; 3Preclinical Paediatric Cancer Evolution, Centre for Cancer Drug Discovery, The Institute of Cancer Research, Cotswold Road, London SM2 5NG, UK; Alejandra.bruna@icr.ac.uk

**Keywords:** neuroblastoma, preclinical, PDX, organoids

## Abstract

Patient-derived preclinical models are now a core component of cancer research and have the ability to drastically improve the predictive power of preclinical therapeutic studies. However, their development and maintenance can be challenging, time consuming, and expensive. For neuroblastoma, a developmental malignancy of the neural crest, it is possible to establish patient-derived models as xenografts in mice and zebrafish, and as spheroids and organoids in vitro. These varied approaches have contributed to comprehensive packages of preclinical evidence in support of new therapeutics for neuroblastoma. We discuss here the ethical and technical considerations for the creation of patient-derived models of neuroblastoma and how their use can be optimized for the study of tumour evolution and preclinical therapies. We also discuss how neuroblastoma patient-derived models might become avatars for personalised medicine for children with this devastating disease.

## 1. Introduction

Neuroblastoma belongs to a group of diseases known as neurocristopathies, a term coined by Bolande in 1974 to describe abnormalities of neural crest development [1]. This embryonal tumour can arise from any part of the sympathetic central nervous system, and over 90% of sporadic cases occur in children under the age of five [2]. The clinical definition of high-risk disease is the presence of distant metastatic spread and/or the presence of MYCN amplification. Clinical presentation in high-risk disease is often with an abdominal mass, arising from the adrenal medulla, with extensive metastatic deposits to the bone marrow, bone, lymph nodes, liver, and skin [3]. Alternatively, neuroblastomas can originate from the sympathetic ganglia and paraganglia in the cervical or thoracic regions, or the retroperitoneal or pelvic spaces. Neuroblastoma is a highly proangiogenic tumour and has a propensity to encase the large vasculature of the anatomical compartment in which it arises [4,5]. 

The finding of MYCN amplification, segmental chromosomal alterations, and histological features of undifferentiated tumour cells with a high Mitosis-Karyorrhexis index are all markers of aggressive disease [6,7,8]. However, the highly variable clinical course of each patient is determined by a multitude of other factors, such as the accumulation of chromosomal breaks and copy number changes [9], gene mutations, and telomere maintenance mechanisms [10]. High-risk tumours have a very poor survival outcome and may either be refractory to upfront treatment or progress despite an initial response to standard therapy. MYCN-driven neuroblastoma accounts for about half of all high-risk cases, and next-generation sequencing has allowed for the identification of further mutually-exclusive subgroups of poor-outcome disease. These include either ATRX (a-thalassemia/mental retardation syndrome X linked) mutations, which can lead to the ALT (alternative lengthening of telomeres) phenotype, or TERT overexpression, as a result of TERT gene rearrangements. Expression of the tyrosine kinase anaplastic lymphoma kinase (ALK) [11] and hyperactivation of the RAS/MAPK [12] pathway is also frequently found in high-risk and relapsing neuroblastomas, although ALK expression itself is not universally associated with a poor outcome [13,14]. 

Early-phase clinical studies open for patients with neuroblastoma are evaluating novel therapeutics, immunotherapies, and combinations [15]. These have all been initiated on the basis of preclinical evidence of efficacy, with an emphasis on the need to demonstrate mechanism of action in vitro and tumour targeting in vivo. However, there remains an unacceptably high attrition rate of 80–90% for compounds to reach clinical practice following phase 2/3 clinical trials, usually due to lack of sufficient efficacy [16,17]. One of the major contributing factors for this in a rare disease such as neuroblastoma is the urgent need for biomarker-led studies in highly relevant preclinical models. Rapid clinical sequencing for paediatric cancer patients is becoming a widespread practice through several different platforms across the developed world [18,19]. To act on the results of molecular analyses requires robust preclinical evidence of the most promising novel therapeutic candidates. Historically, the study of neuroblastoma has relied upon the data provided by immortalized cancer cell lines. However, it is widely acknowledged that these lines have grown in artificial cell culture conditions and consequently drifted so that they no longer resemble the originating tumour, and therefore they have limited clinical predictive power [20]. Alternatively, patient-derived cancer models are created from the in vivo implantation into immunodeficient mice (or in vitro culturing) of tumour cells or fragments of tumour taken directly from a patient with cancer. These models should accurately reflect the molecular make-up of the patient’s tumour and therefore provide a more representative model for the study of tumour growth and response and resistance to therapeutics. In rare diseases, and especially in paediatric malignancy, where repeated invasive procedures to obtain tumour samples can be difficult to justify, patient-derived models are of even greater importance for allowing us to gain insight into the biology of the disease and improve drug development. 

Most interestingly, a recent study using 1000 adult cancer patient-derived xenograft (PDX) models demonstrated that drug combinations which had been pursued clinically after traditional cell line research were not so effective in the PDX, which concurred with the disappointing clinical trials [21]. Also of note, the US National Cancer Institute recently decided to replace its panel of 60 human cancer cell lines (NCI-60), which had been used for 25 years for drug screening, with a panel of 60 PDX which come with extensive details about the clinical course of the tumour from which they were derived [22]. This move set the gold standard for cancer research to include models closely matched and most relevant to the human disease counterpart.

Of course, patient-derived models have their own drawbacks. Highly heterogenous malignancies, such as high-risk neuroblastoma, from which a single biopsy is often taken for diagnosis, do not represent the physical and molecular structure of the entire tumour. Furthermore, the tumour evolutionary response to therapy plays a significant role in intra-tumoural heterogeneity and can lead to the expansion of drug-resistant clones. In fact, high genetic diversity after chemotherapy is unequivocally associated with a poor outcome in neuroblastoma [23,24]. Therefore, establishing preclinical models from patient tumour tissue and subsequently using those models for drug testing, must be carried out with the caveat that artificial selection of tumour tissue will have occurred. Genetically-engineered models (GEMM) of neuroblastoma are also used in translational research and provide an essential means to investigate the basic mechanisms of neuroblastoma initiation and progression, coupled with an opportunity to undertake preclinical drug testing in a heterogeneous tumour cell population within an immune-competent host. This review, however, focusses on the development of, and the potential for patient-derived models for the study of neuroblastoma.

## 2. Ethical Considerations for the Development of Patient-Derived Xenografts

Pioneering experiments to establish neuroblastoma xenografts using patient tissue were begun over thirty years ago. This work provided early evidence that patient tumour tissue implanted in nude mice lead to the growth of preclinical neuroblastomas, which shared many of the molecular phenotypes of the original patient tumour and related cell lines [25]. These early models also demonstrated the potential for xenografts to model the sensitivity to therapy and scheduling of treatment protocols for neuroblastomas [26,27]. In the intervening years, PDX models have become an extremely valuable resource in cancer research. With the stakes so high for their influence on the next generation of clinical trials, it is imperative that standards for their generation and maintenance are kept accountable.

The minimum reporting requirements for PDX have been published following a review of the standards set by independent PDX model consortiums [28]. These standards are proposed to be universally implemented by all institutions working to develop relevant PDX models. They include firstly, accurate reporting of clinical details from the donor patient, including crucially, their consent to share tissue and data. Secondly, details of the pathological diagnosis of the submitted tissue and whether the patient had received any treatment. 

The sponsors of patient tissue collection studies, which aim to create PDX, face complex ethical considerations. Informed consent and, whenever possible, assent from the child need to be obtained as soon as possible, including consent for tissue to be taken and utilized for research and for data to be collected. There is broad discussion on the need to confirm consent in former patients who become adults. Contacting individuals to confirm consent represents a logistical challenge and a significant cost. The approach taken by ethics committees has been heterogeneous, from waiving the need to re-consent, to making it mandatory. Making confirmation of consent mandatory can prevent the use of important research material and delay life-saving discoveries. The decision from the ethics committees stems from the definition of human research subject (some have argued that if the samples are completely anonymized, consent is no longer needed) and of parental responsibility. Most ethics committees have taken a cautious approach of recommending an attempt at confirming consent [29,30]. Of note, when adult survivors of children cancer were asked if they thought it was important for consent to be confirmed, their responses varied greatly, but all granted consent to use their historical samples for research [29]. In our experience, only a minority of patients express concerns about the use of animals in research. 

In the consent process, it is important to illustrate the challenges of tissue prioritization, which arise from an increasing number of tissue studies. While undoubtedly the priority has to be given to diagnostic procedures, it is harder to prioritize among research protocols. As the generation of PDX requires fresh tissue, the pathologist faces the difficult decision on whether there is available tissue for PDX, in particular in case of small biopsies, sometimes prior to having the diagnosis. The potential consequences include the inability to achieve the diagnosis (and the potential need to repeat an invasive procedure), lack of tissue to perform molecular tests (such as next generation sequencing, which could in turn identify therapeutic targets), or lack of residual material to be used in the future for screening for clinical trials. The technical improvements in molecular tests (which require less and less tissue and can often be performed on fixed tissue) and the widespread use of genome-wide investigations (as opposed to single-gene testing) to determine eligibility to treatment with targeted agents will eventually facilitate this decision. Although some authors have attempted to formalize their tissue prioritization protocols [31,32,33], this is a dynamic balance, and we can anticipate that PDX will move up in the priorities list, as their role in treatment decision making becomes broader. 

The expectation for any successful PDX to be used as real-time “avatars” for personalised treatment planning should be discussed in advance. There are now a handful of examples of paediatric PDX models which have been used as a surrogate for testing conventional and novel therapeutics. As the time taken to establish these models is shortened, it is inevitable that data generated in vivo could influence clinical decision making at an individual patient level [34,35]. A recent notable study aimed to find out the acceptability for the use of PDX to personalise patient care amongst adult and paediatric cancer survivors, parents of childhood cancer survivors, and control groups who had not experienced cancer treatment first-hand [36]. This study found that even with the longer wait-times for PDX generation and testing, cancer survivors and their parents were generally willing to consider the use of PDX to guide individual therapy. 

The collection of patient data to accompany the tissue collected for research must be taken with full consent, anonymised to the investigators, and securely stored according the data protection laws within each country. In the UK, the General Data Protection Regulation (Data Protection Act 2018) ensures good practice in data protection to support the innovative use of data, while the Human Tissue Act (2004) regulates the removal, storage, use, and disposal of human tissue. 

It is of the greatest importance that procedures and regulations are standardized globally, in order to compare results of trials and to be able to share data broadly, to maximize their impact. The Global Alliance for Genomic and Health (https://www.ga4gh.org/, accessed on 25 March 2021) is an example of such an attempt. Unfortunately, the data protection regulations remain different in the USA, Europe, and the UK. 

To comply fully with these legislative acts requires a complete collaborative endeavour between oncologists, research nurses, pathologists, surgeons, and researchers, which can only come about with extensive prior planning. 

For any in vivo study, due consideration of the ethical and experimental reporting guidelines for animal studies is also of upmost importance. The ARRIVE guidelines (Animal Research: Reporting of In Vivo Experiments) were first published in 2010 and have recently been updated to help improve the consistency in their implementation [37,38]. The key aim of these guidelines is to ensure reproducibility of in vivo studies in order to provide the best possible evidence-based preclinical work to guide clinical studies, whilst maintaining public confidence in science involving animal research. The principles of replacement, reduction, and refinement (NC3Rs: National Centre for the Replacement, Refinement, and Reduction of Animals in Research) are a core element of any biological research. Wherever possible, new approaches should be sought to avoid the use of animals in addressing important scientific questions. In the case of patient-derived models, we discuss further on the advancements in the use of 3D patient cultures and how these can be a valuable alternative or intermediate step before animals are used. When animals are required, their numbers should be limited to those needed to achieve robust and reproducible results. Finally, the welfare of animals used in scientific research should be under continuous review, so that refinements in line with up-to-date technologies can support the impact of welfare on scientific outcomes. 

## 3. Technical Considerations for the Development of Patient-Derived Xenografts 

Referring to the minimum reporting requirements for PDX model generation is a good starting point for considering the technical aspects of establishing PDX [28]. Aside from the patient information described above, information regarding the creation of the PDX, including the mouse strain and source and whether the patient tissue was engrafted as a solid piece or might have been dissociated into single cells, should be recorded. Testing should be undertaken to demonstrate that any successful PDX model is characterised to ensure that the tumour represents the matched human sample and is not of mouse or Epstein–Barr virus (EBV) origin. Evidence that the PDX model can be passaged in vivo and that the model can be rescued following the freezing down of live tissue or cells is also essential.

The choice of mouse strain is an important early consideration. Previously nude mice were the most widely available animals; however, the success rate for engraftment of PDX was limited using this strain [39]. NOD scid gamma (NSG) mice, which can have a longer life span than Nude mice, are now more widely available, and their use has improved the engraftment rate. This is especially true for neuroblastoma, as the latency period for this tumour to grow as a PDX can be up to one year [40]. Although NSG do not develop spontaneous thymoma, their incidence rate of EBV-associated B-cell lymphoma has been reported to be as high as 32% [41]. Without a T-cell response NSG are vulnerable to T-cell controlled infections [41,42,43]. Most people are infected with EBV, and therefore, EBV-associated oncogenesis is a risk, as it is driven by T-cells, which may be engrafted with the patient tumour sample [44,45]. Before using a novel PDX in further experiments, such as for personalised medicine, this possibility should be discounted. Xenogenic graft versus host disease has also been described in neuroblastoma PDX, where mice were engrafted using cells from either a lymph node or bone marrow metastatic site [46]. To overcome this, it is highly recommended to undertake a lymphocyte depletion step of the sample prior to in vivo implantation.

There remains some controversy regarding the overall evolution of PDX, with some arguing that PDX will evolve in a mouse-dependent manner, whilst others report that human tumours and PDX tumours evolve in parallel [47,48]. The clonal evolution of PDX tumours through in vivo passaging has been studied primarily in PDX models of adult cancer. Deep-genome and single-cell sequencing of breast organoids suspensions engrafted into mice illustrated the large variety of clonal selection from minor (<5%) to moderate or polyclonal engraftment, which was reproduced across independent grafts of the same starting population [49]. Furthermore, the clonogenicity of an extensive biobank of breast cancer PDX was characterised using PyClone, which is a statistical model allowing Bayesian clustering of clonal somatic mutations and estimation of their cellular prevalence [50,51]. The greatest shift in clonal selection was observed upon initial engraftment, but subsequently, selection was minimal through serial in vivo passaging of tumour pieces. Examination of paired PDX samples within this study from both primary and metastatic patient samples revealed that clonal clusters shared by both the originating primary tumour and metastatic samples had stable cellular prevalence across passaging. This finding strongly endorses the notion that the dynamics of clonal evolution between the original patient samples and the PDX and during in vivo passaging of the PDX, are not stochastic. If clonal dynamics could be similarly studied in neuroblastoma PDX, it would lend enormous weight to the support of these models to preclinical studies. 

The heterogeneity of PDX tumours is not only governed by cell-autonomous clonal selection but through the influence of the tumour-associated stroma. Quality control of PDX tumours which elucidates the proportion of mouse stromal cells replacing the original xenografted tumour should also be part of any PDX characterisation [52]. Without this knowledge molecular analysis of a PDX tumour at genomic, epigenomic, and transcriptomic levels will be compromised. Computational tools exist to classify sequencing reads belonging to two different species [53,54] and a recently developed more rapid in silico combined human-mouse reference genome can be utilized in the computational alignment set [52]. It is essential to monitor the lineage of PDX passages closely in order to define the points at which a human PDX has been replaced by a murine tumour. 

## 4. The Development and Refinement of Patient-Derived Models for Neuroblastoma

### 4.1. Patient-Derived Neuroblastoma Xenografts

PDX have the potential as models for a variety of applications, including the validation of molecular targets, preclinical assessment of new compounds, the modelling of drug resistance, and the study of tumour evolution (Figure 1). In addition, personalised PDX avatar models could be utilized to guide therapy for an individual patient under the right circumstances. The barriers to this success in neuroblastoma to date have been the poor engraftment rate, the long latency to tumour formation, and the high cost of initiating and maintaining this work over long periods of time. The methodological considerations for researchers include whether to use fragments of tumour to engraft or to dissociate the tumour material before engraftment, and whether to opt for subcutaneous engraftment or to use a neuroblastoma-associated orthotopic site, most commonly the adrenal [46,55,56,57,58,59,60]. 

Efforts to improve the engraftment rate of neuroblastoma have led to the development of several examples of orthotopic models. One of the first reported successful orthotopic PDX neuroblastoma model was of a post-chemotherapy high-risk patient tumour, dissociated and then engrafted into the para-adrenal space of an NSG mouse [60]. After six months, a tumour had developed, and metastases were identified in the liver. At first, in vivo passage tumours grew more rapidly, which is consistent with published data in other tumour types [47]. This can be explained by mouse specific evolution, where there is selection of tumour cells which have adapted to grow in the mouse or due to high cellularity of the tumour harvested at passage zero. Molecular and genomic characterisation of the patient tumour tissue, the primary adrenal xenograft, the liver metastases, and the passaged tumours revealed a high level of molecular concordance, with maintenance of the chromosome break points. Dissociated cells from the orthotopics were used to undertake drug screening in vitro to demonstrate the feasibility of this approach. Notably, it was found that the cells from the orthotopic model were less sensitive to chemotherapy agents than the cultured neuroblastoma cells included in the same assay. It has also been described that orthotopic PDX models of neuroblastoma are more likely to demonstrate metastases when compared to subcutaneous models [56] or to cell-line-derived orthotopic tumours [57]. The immune infiltration of intra-adrenal models of Th-MYCN GEMM tumour cells has been compared to subcutaneous models, finding that tumour-associated immunosuppressive macrophages were more abundant in the orthotopic model [61], suggesting that orthotopic PDX models might also be superior models of the tumour microenvironment.

Furthermore, a recent article directly compared the time to engraftment between a subcutaneous and orthotopic adrenal neuroblastoma tumour xenograft [46]. They found that engraftment was faster at the adrenal site, and also that engraftment rate can be improved by increasing the number of cells implanted. In addition, the establishment of a PDX from residual cells following clinical cytogenetic analysis was the first of this kind reported and interestingly was successful in two cases where the tumour material engrafted in parallel was unsuccessful. Molecular correlation between the models established from the cytogenetic residual cells and the original patient tumour was good, and drug sensitivity was similar between the PDX and dissociated cells from the patient sample. This article serves as a good example that the opportunity for establishing PDX can be taken from many different sources of patient material. 

Optimising the imaging of orthotopic PDX models is also an important consideration. Ultrasound (US) imaging is widely cited for the monitoring of PDX growth. However, a comprehensive comparison of imaging to compare magnetic resonance imaging (MRI), US and luciferase signal between the Th-MYCN GEMM and several adrenal orthotopic models of neuroblastoma cell lines found that MRI was the most accurate in size measurements of the tumours [62]. Bioluminescence has been widely adopted as a relatively rapid method to monitor in vivo growth of tumours, especially when using orthotopic models that cannot be measured by callipers. However, it must be considered that the transfection of cells in vitro results in the expression of novel tumour antigens [61]. These could change the immunogenicity of a tumour grown subsequently in vivo from these cells, and therefore it is advisable to follow up animal studies using untagged tumours. Also of note, in highly necrotic tumours, there is a lack of correlation between tumour size and bioluminescence signal [63]. 

Fluorine 18 fluorodeoxyglucose (FDG) positron emission tomography (PET) can performed clinically for children with neuroblastoma, whose tumours are not amenable to metaiodobenzylguanidine Iodine-123 (MIBG) scintigraphy, because of lack of MIBG uptake by the tumour [5]. Either of these investigations are essential for the detection and follow-up of metastatic disease. FDG-PET was used alongside MRI to monitor the metastatic behaviour of orthotopic adrenal PDX models of neuroblastoma [57]. The models used in this study were generated by the para-adrenal engraftment of a fragment of human neuroblastoma tissue, rather than dissociated cells. The authors demonstrated that both MRI and FDG-PET could be successfully used to track the intra-abdominal growth of these PDX tumours, which could not have been prelabelled for bioluminescence. 

A criterion for the successful establishment of a PDX model, is the ability to continue to passage the tumour successfully in vivo [28]. However, this approach could easily impact the clonal evolution as the tissue bank is expanded through generations of mice. This effect was studied in orthotopic adrenal neuroblastoma PDX up to passage eight of sequential tumour fragments, in order to monitor the genetic, transcriptional, and phenotypic stability over time [72]. Within this study, orthotopic PDX were generated by the para-adrenal engraftment of undissociated tumour fragments, both at the beginning of the study directly from the patient tissue and through in vivo passage. In this way, the authors aimed to avoid artificial clonal selection through the act of dissociating cells. Reassuringly, not only did the genetic, transcriptional, and phenotypic status remain preserved, but gene signatures also remained stable over the two years of the study and correlated with patient outcome. 

The extensive intra-tumoural heterogeneity of clinical neuroblastomas makes the use of neuroblastoma PDX as avatars for personalised patient medicine very challenging. In PDX models of other cancer types, avatars have been used successfully to personalise the therapeutic approach for individual patients, through the assessment of both conventional and novel therapeutics, in co-clinical trials that can be performed whilst the patient is in active treatment [73]. This opens up the opportunity to monitor the sensitivity or innate or acquired resistance of the tumour to treatment and the identification of biomarkers of response. However, spatial and temporal heterogeneity is a barrier to the success of this approach for neuroblastoma. This heterogeneity has recently been subtyped into four evolutionary trajectories that may be followed simultaneously by a single tumour [23]. Mapping of 250 regions in 54 childhood cancers to perform whole-genome analysis led to the construction of a personal evolutionary ideogram for each patient, reflecting the distribution of genetically distinct cancer cell populations over anatomical space and treatment time. The four trajectories revealed were the following (with death from disease associated with types 3 or 4):Subclones with very few mutations confined to a single tumour regionStable coexistence, over vast areas of clones characterised by chromosomal number aberrations (this was demonstrated by half of the neuroblastomas in the study)A clone with a driver mutation or structural rearrangement emerges through a clonal sweep to dominate an anatomical region (again, this was shown in half of neuroblastomas in the study)Local emergence of a myriad of clones with TP53 inactivation.

This work also found that metastatic recurrences tended to be genetically distinct from the primary tumours, emphasizing the case for resampling at relapse. Intra-tumoural heterogeneity and cancer evolution of course has major implications for drug discovery [74]. Although very difficult, PDX models have the potential to guide treatment for high-risk malignancies which may show huge variety in clonality. Indeed, this has been already studied using multiple PDX from multiple biopsies of a single patient with a high-risk neuroblastoma [72]. The tumour was found to have a very diverse transcriptional, proteomic, and phosphoproteomic profiles, suggesting significant spatial intra-tumoural heterogeneity. For tumours such as this, it may be possible to perform drug screening on all the PDX models generated. This would also allow tracking of subclonal chemotherapy resistant populations. Alternatively, PDX might be created from heterogeneous patient-derived cell cultures or liquid biopsies, assuming that these tumour cells would capture the intra-tumoural diversity of the patient tumour. Evidence from adult cancer PDX models suggests that it is possible to predict drug responses in a relapsing clinical tumour from analysis of the PDX created from early disease. If the time to neuroblastoma PDX generation can be significantly shortened, it is exciting to envisage that this anticipatory personalised approach might be possible for paediatric cancer patients

### 4.2. Application of Patient-Derived Neuroblastoma Xenografts

The Paediatric Preclinical Testing Consortium (PPTC) is currently the largest repository of PDX models of high-risk paediatric malignancies, with which screening of novel therapeutics is undertaken to help accelerate their translation to clinical trials for children with cancer [75]. The non-brain-tumour models represented in this cohort were grown subcutaneously, and comparison of each model to the human tumour of origin revealed high concordance of somatic mutations, copy number alterations, RNA expression, gene fusions, and activated signalling oncogenic pathways. This study supports previous reports of smaller panels of paediatric PDX models, in that there should be high confidence in their predictive value for drug screening [59,86,87,88]. This confidence can be enhanced when cohorts of PDX models from a particular molecular subtype are employed for biomarker-driven preclinical studies. In Europe, the Innovative Therapies for Children with Cancer—Paediatric Preclinical Proof-of-Concept Platform (ITCC-P4) is a further example of a PDX repository of high-risk paediatric malignancies. Both the PPTC and the ITCC-P4 work in partnership with the pharmaceutical industry to prioritise novel agents for screening. 

To mention all the preclinical studies for neuroblastoma which have incorporated PDX models is beyond the scope of this article, but several stand out as examples of how PDX can strengthen the evidence for novel therapeutics and combinations to enter clinical studies.

ALK is the first directly targetable tyrosine kinase to be identified as a driver oncogene in neuroblastoma. PDX models of ALK mutant neuroblastoma have been pivotal in recommending particular ALK targeting small molecules for clinical studies. Of particular note, the Felix neuroblastoma PDX, which was established post-mortem from the blood of a patient who died of *ALK^F1245C^ MYCN*-non-amplified high-risk neuroblastoma, has been part of several publications to highlight the potential of synergy between ALK inhibition and chemotherapy [76], the resistance to ALK inhibition mediated by Pim1 [77], and the study of novel ALK-antibody conjugates [78]. The Felix neuroblastoma PDX harbours the third most common ALK mutation found in neuroblastoma, and it exhibits de novo resistance to the first generation ALK inhibitor, crizotinib, which mirrors that seen clinically for this subgroup of patients. 

Prior to committing to in vivo experiments, it is helpful to establish evidence of efficacy using in vitro screening with either cell lines or ideally patient-derived in vitro models or ex vivo PDX tumour cells. For example, the triple PIM/PI3K/mTOR inhibitor IBL-302 was used to treat an N-myc expressing neuroblastoma PDX in vivo, in combination with reduced-dose Cisplatin chemotherapy, following extensive in vitro screening [79]. A panel of *MYCN*-amplified neuroblastoma organoids was employed to run a high-throughput screen, selecting the inhibitor of kinesin spindle protein, ARRY-520, as an agent to cause reduced organoid viability [64]. This compound was then further tested preclinically in neuroblastoma N-myc PDX in vivo models, taken from a patient at diagnosis and at treatment-resistant relapse. Interestingly, survival was prolonged further in the adrenal orthotopic PDX model, in comparison to the subcutaneous model.

### 4.3. Advancing Neuroblastoma Immunotherapy Using Patient-Derived Models

Progress in the treatment of high-risk neuroblastoma was accelerated in recent years through the addition of immunotherapy to target minimal residual disease [80]. The current clinical protocol allows for the administration of interleukin (IL)-2, granulocyte-macrophage colony-stimulating factor (GM-CSF), and anti-disialoganglioside (anti-GD2) antibody with the differentiation agent cis-retinoic acid. Although this combination has significantly improved event-free survival, there still remains a subgroup of patients who relapse whilst receiving this therapy or who are unable to tolerate the treatment-related toxicity of IL-2.

An elegant study by Barry et al (2018) aimed to recapitulate this clinical scenario preclinically using neuroblastoma PDX [81]. This study was undertaken to investigate whether the adoptive transfer of activated natural killer (NK) cells could improve the response to anti-GD2 immunotherapy through the induction of antibody-dependent cellular cytotoxicity (ADCC) in consolidation therapy for children who have had surgical resection of their tumours. An orthotopic kidney neuroblastoma PDX tumour was allowed to grow, before the animal underwent macroscopic resection of the kidney. Animals were then treated with a combination of the GD2 antibody, dinutuximab with activated NK cells, versus control groups, and there was a significant reduction in the incidence of liver and bone metastases, improving the survival in animals which had received the combination. 

In a separate study aiming to improve the outcome for children who do not respond to current immunotherapy protocols, neuroblastoma PDX-bearing CD1-FOXn1^nu^ mice were utilized, as although they are immunocompromised athymic animals, they retain fully functional endogenous NK cells, albeit at reduced numbers [82]. Use of this strain for neuroblastoma PDX allowed the study specifically to compare NK cell-mediated ADCC due to either IL-2 or IL-15 activated NK cells. IL-15 was of particular interest as an alternative cytokine, as it binds to the same receptor on NK cells as IL-2 and is being successfully clinically developed in adult cancer patients [83]. This first-of-its-kind study of IL-15 in an established neuroblastoma PDX model provided compelling preclinical data that IL-15 should be evaluated in future clinical studies, at least to compare its tolerability against that of IL-2 in paediatric patients. 

Humanized mice, which are engrafted with components of the human immune system, have become extremely valuable for the study of immuno-oncology, as they allow study of the interaction between patient-derived tumour models and the immune microenvironment [84]. Most recently, an orthotopic PDX model of neuroblastoma which supports human NK cell development was described [85]. The extensive characterisation of this model allowed for the conclusion that human NK cells could be therapeutically modified to induce antibody-dependent cell-mediated cytotoxicity. Interestingly, activated NK cells were reduced in this PDX model, which might be a tumour-associated phenomenon that has already been reported in immune profiling studies of neuroblastoma patients.

### 4.4. Patient-Derived Neuroblastoma Zebrafish Models

The engraftment of patient tumour samples into zebrafish for rapid preclinical drug screening is gaining in popularity and has the potential to bridge the gap between expensive, slow murine studies, and clinical drug development. Zebrafish modelling has been undertaken successfully for melanoma, rhabdomyosarcoma, paediatric brain tumours, and, more recently, neuroblastoma [89,90,91,92]. Several attributes of the zebrafish make them particularly suitable for this type of study: firstly, the transparency of the zebrafish embryos and larvae which can be prolonged through the use of pigment inhibitors; the lack of a fully functional adaptive immune system in the first weeks of life; the availability of genetically-modified strains with defined immunodeficiency; and that xenografted models only required a few hundred cells for successful tumour development which can be visualised at the early stages to monitor vascularisation [93]. Histone deacetylase (HDAC) inhibitors were combined with chemotherapy in a neuroblastoma patient-derived zebrafish model. This model was created through the injection of tumour cells into the yolk sac of zebra fish larvae to monitor response to drug treatment or the injection of tumour cells into the perivitelline space to undertake analysis of tumour cell dissemination [89]. The xenografted tumours grew within a very short timeframe in this model and therefore allowed rapid analysis of therapeutic intervention. However, short-comings of this approach include the difficultly in translating these pharmacokinetics across to human, and the limited tumour xenograft material, making pharmacodynamic studies challenging. 

### 4.5. Patient-Derived Neuroblastoma 3D In Vitro Cultures

Whilst successful PDX models provide an unparalleled model of patient tumour diversity, the high cost and time taken to deliver these models, in addition to weighing up the benefit to human health versus the harm to the animals involved, are important considerations. The 3Rs principle of replacing, refining, and reducing animal use particularly lends weight to the argument for the extended use of patient-derived 3D in vitro cultures, as an alternative to address the tumour biology and translational study of neuroblastoma [94]. There is considerable variety in how these lines can be derived. Growing neuroblastoma tumour tissue-derived lines in low attachment conditions such as spheroids, with or without the addition of Matrigel, is widely practised. However, the derivation of patient-derived, self-renewing tumour organoids, remains a challenge for non-epithelial-origin malignancies, such as neuroblastoma. Using in vitro systems for the testing of therapeutics prior to in vivo study is well established, but sophisticated bioengineering has the potential to increase the relevance of 3D in vitro studies to the study of neuroblastoma biology.

The generation of primary neuroblastoma spheroid cultures in neural stem cell serum-free media can enrich for tumour-initiating cells (TICs), which reflect the genetics and phenotype of the patient tumour and can be xenografted into immunocompromised mice [65]. The absence of serum in the media is considered an essential element to allow for this close resemblance between the spheroids and the patient tumour [66]. A standardised approach to establishing spheroid culture of neuroblastoma patient samples in serum-free neurosphere culture was found to be successful in 55% of cases in one study, in which cultures were established from chemotherapy-naïve primary tumours, postchemotherapy surgical samples, and primary needle biopsies [67]. Serial sphere passaging is also a well-established assay for self-renewal, and the ability of spheroids to survive repeated passage suggests enrichment for TICs. This was elegantly demonstrated by Coulon et al, who also showed that patient-derived spheroids had enhanced tumourigenicity following orthotopic adrenal implantation and increased resistance to the chemotherapy agents doxorubicin, etoposide, and cisplatin, compared to their non-spheroid counterparts [68]. 

Tumour-derived organoids should maintain the genetic heterogeneity of the primary tumour tissue over time, and the predictive value of therapeutic testing for individual patients has been very promising [69]. The first organoid bank for paediatric kidney cancer contains tumour and matching organoid cultures for more than 50 children with Wilms and non-Wilms tumours [68]. Significantly, this biobank includes organoid models of malignant rhabdoid tumours which are the first described such tumour models of nonepithelial origin. Not only did extensive characterisation of this biobank reveal that the organoids recapitulated patient copy number alterations and mutational signatures but also that patient-specific drug sensitivities were retained. For neuroblastoma, Fusco et al. have recently described the development of a panel of six neuroblastoma patient-derived organoids [70]. Again, the organoids retained the genetic and phenotypic features of the original tumour from which they were derived and were superior to matching spheroid culture, as only the organoids, grown in cell suspension embedded in Matrigel, could reproduce the tumour morphology and architecture. Furthermore, neuroblastoma patient-derived organoids have been incorporated into a study identifying the gene TBX2 as a component of the neuroblastoma core transcriptional regulatory drive [71]. Within this paper, neuroblastoma organoids were utilized to strengthen the evidence for observed synergy between CDK7 and BET bromodomain inhibition to circuit break this transcriptional addiction.

Neuroblastoma organoid cultures also have the potential to be generated from embryonic stem cells or induced pluripotent stem cells [95]. Human models of neuroblastoma can be generated from the neural crest, which is a multipotent embryonic cell population giving rise to various cell types, from melanocytes, neurons, and glia cells originating from the cranial neural crest to the sympathetic neurons and neuroendocrine cells originating from the truncal neural crest. It is possible to differentiate human pluripotent stem cells into subtypes of neural crest cells in vitro by utilizing methods to manipulate the signalling of markers which have been characterized through the study of axial neural crest differentiation in different species [96,97,98]. As a result, human cell models of truncal neural crest stem cells can be generated to study the biology of neurocristopathies, including neuroblastoma. The isolation of trunk neural crest cells from Trp53-null mouse embryos, which were then transformed by N-Myc, can form tumours subcutaneously with 100% penetrance in nude or C57Bl/6 mice [99]. The tumours had a mixed phenotype but predominantly exhibited markers of neuroblastoma, with metastases observed in the lymph nodes, liver, and lungs. Although this study was carried out in mice, the principle of tumour development from the transformed neural crest gives hope that human-derived neural crest models may be generated in vivo.

Finally, whilst in vitro 3D tumour cultures may seem to be an unlikely model for the study of immunotherapeutics, recent advances to generate organ-on-chip short-term spheroid cultures with accompanying microfluidic culture have the potential to provide an alternative preclinical model for novel therapeutics [100]. Microfluidics refers to the ability to manipulate a small volume of fluid in a network of microchannels that can be contained on a chip. This unique experimental platform has already been used to evaluate the response of murine and patient-derived 3D tumour cultures to immune checkpoint blockade [101]. In this case, it was found that ex vivo tumour samples retained lymphoid and myeloid cell populations, and the profiling of secreted cytokines in this model matched what was seen in in vivo experiments with the same agents. Not only does such work reduce the dependency on animal experiments for characterisation of the tumour microenvironment but it also addresses the problems of the time taken to establish murine models and the lack of native immune response in immunodeficient models. Further, a model of patient-derived glioblastoma-on-a-chip was used to describe the immune-modulatory mechanisms that are associated with resistance to PD-1 inhibition, thereby setting the stage for personalised immunotherapy screening in this group of high-risk patients [102]. Similarly, dysregulation of PD-1 expression has been described in neuroblastoma; therefore, novel approaches to predict which patients might benefit from PD-1 inhibition would be of great value. 

## 5. Conclusions

High-quality preclinical modelling of the diverse clinical behaviour of neuroblastoma is hugely challenging yet essential to investigate the evolution of these tumours and to deliver an optimal predictive value for the study of novel therapeutics. Orphan drugs for rare diseases, such as neuroblastoma, take an especially long time to reach patients. Biomarker-led studies are generally more successful, which emphasizes the need for a broad range of preclinical neuroblastoma models which represent the major molecular subtypes. In such a rare disease, this can only be achieved through the collaborative efforts of initiatives such as ITCC-P4 in Europe, and the Paediatric Preclinical Testing system in the US. 

Ethical protocols to collect patient samples should be established following considerable discussion of how the study fits into the national ethical legal obligations whilst considering the practicalities of local clinical and laboratory services. We would recommend that once a laboratory is approved for the collection of patient samples, that the minimum reporting requirements for PDX are followed and that a biobank is established to freeze down PDX tumour material in large batches from early passages. The PDX samples should be rigorously tested to guard against the inadvertent development of murine tumours. As tumours intrinsically evolve, when PDX samples are used in further experiments, for example in preclinical trials, they should be double-checked for expression of the relevant targets or biomarkers to ensure accurate interpretation of data. 

Ultimately, will the standard panels of immortalized human neuroblastoma lines be replaced by well-characterized in vivo and primary cultures of human tumours, that better reflect the genomic make-up of the human tumour from which they were derived? As more researchers adapt their work to the use of these superior models, it is becoming clear that to move novel therapies from preclinical to clinical, we require robust evidence of efficacy from the most relevant models. The remaining challenge is to drastically improve the engraftment rate and time to engraftment for neuroblastomas as PDX. If this is possible, then the potential of using PDX models as in vivo avatars for children with high-risk disease could change the landscape of personalised medicine for this devastating disease. We envision an improvement of the preclinical setting by developing and validating an integrative pipeline of ex vivo and in vivo studies in patient-derived models as cost-effective intermediate strategies in drug development, which aim to increase the chances of identifying novel therapeutics predictive of patient’s drug responses.

## Figures and Tables

**Figure 1 jpm-11-00248-f001:**
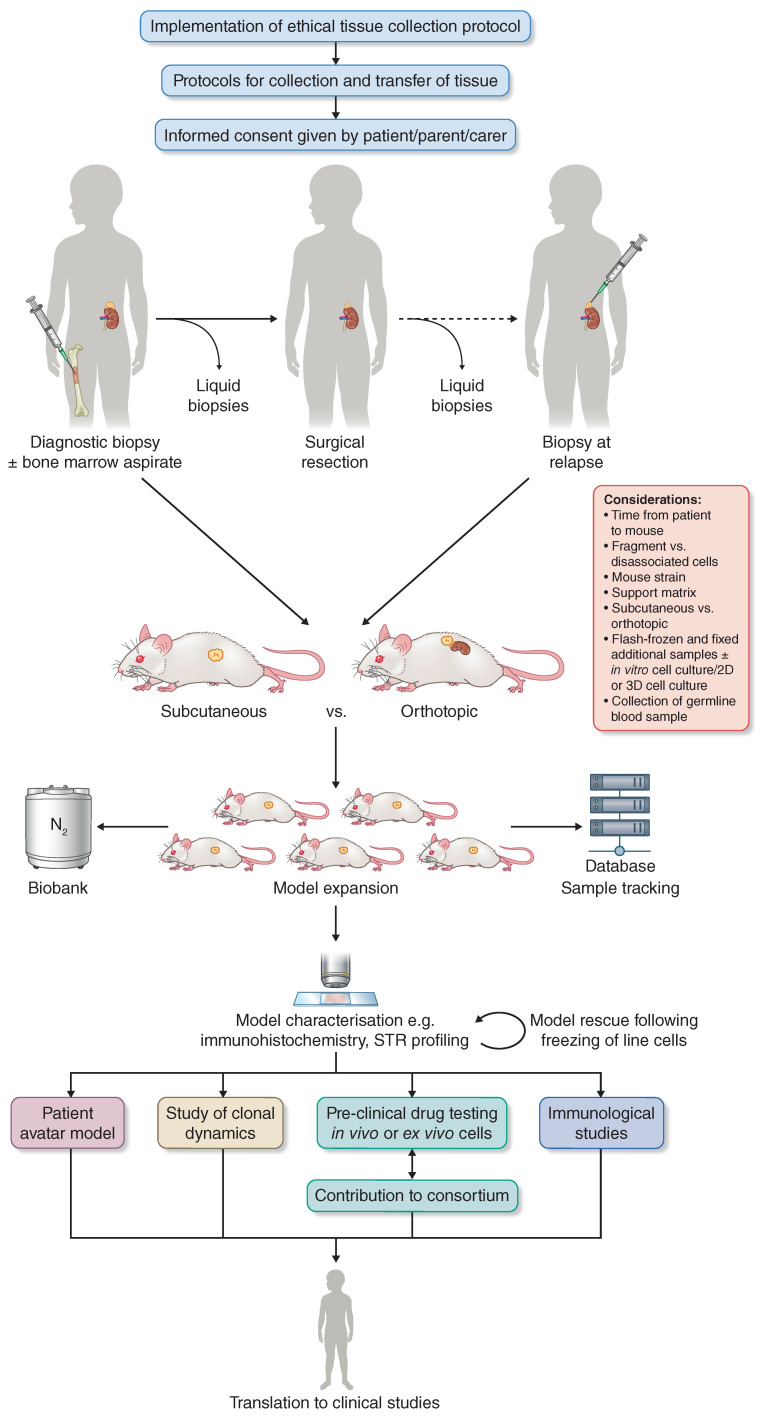
Generalised schematic for the implementation, establishment and potential use of a neuroblastoma PDX biobank. The initial step is to gain local or national ethical approval for a patient tissue collection study, with strong links to clinical research teams in order to acquire the informed consent of patients and/or their careers. Neuroblastoma tumour tissue can potentially be collected following a diagnostic biopsy, at surgical resection, and if further biopsies are taken, upon disease relapse. Additional samples may be taken from liquid biopsies, especially including bone marrow aspirates. A dedicated laboratory team should provide expertise in the engraftment of fresh tissue into immunocompromised mice, with or without the parallel establishment of patient tissue as an attached/spheroid/organoid culture [64,65,66,67,68,69,70,71]. There are multiple considerations at this step (site of engraftment [46,55,56,57,59,60,72] strain [39,40]), and experimental protocols should be guided by the ultimate aims of the study. Once a P0 PDX tumour is growing, the model should be expanded in vivo, with the establishment of a biobank and database for mouse and human sample tracking [28]. Extensive model characterisation should be undertaken at each in vivo passage to characterise molecular concordance with the patient tumour and exclude the possibility of murine EBV-associated lymphoma [41,42,43,44,45,52,72]. At this point, the possibilities for utilisation of a successful neuroblastoma PDX model are far-reaching and include the study of clonal dynamics [47,48,49,50], preclinical drug testing [34,35,72,73,74,75,76,77,78,79], and immunological studies by using humanized mice or switching the mouse strain for further engraftment [80,81,82,83,84,85].

## Data Availability

Not applicable.

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
