# Peer review of "The Promise of Patient-Derived Preclinical Models to Accelerate the Implementation of Personalised Medicine for Children with Neuroblastoma"

_jpm, 2021, doi:10.3390/jpm11040248_

Round 1

Reviewer 1 Report

This is a thoughtful review of the several different kinds of patient-derived, individualized models of neuroblastoma and their potential for development as either drug candidate screening techniques or personalized medicine therapy selection methods.  It is generally well-written but presents more specific examples and facts than evaluation and analysis.  The work would benefit from inclusion of a table that compares the methods described and gives their benefits and shortcomings in areas such as potential for high throughput, degree to which they mimic the original tumor from which they came, cost, durability and potential for dissemination, etc.  In addition, the authors briefly discuss the ethics and legal aspects of sharing such models.  However, they also talk about the relative rarity and the heterogeneity of neuroblastomas, and infer from this the need for multicenter and international consortia to make therapeutic development viable and generalizable.  Given this, they should discuss or at least mention such things as the need for some kind of standardization within and among nations of patient/family consent procedures, need for re-consent once living former patients from whom specimens come reach the age of majority, and mechanisms for storage and dissemination of model systems.

Author Response

Dear Sir,

Many thanks for your review of our manuscript. Herein follows a point-by-point response to your comments:

  1. The review would benefit from the inclusion of a table.
    • The second reviewer requested a figure to summarise the points you have also mentioned.  Upon drafting both a table and a figure, it became clear that a figure would be the optimal way to summarise and reference the important considerations both reviewers mentioned.
  2. The review should discuss the need for standardisation, re-consent, mechanisms for storage and dissemination.
    • Thank you for raising these points.  We have extended our section discussing the ethics around sample collection to include these aspects.

Yours sincerely

Dr Elizabeth Tucker

Reviewer 2 Report

This manuscript summarized the current progresses of establishing preclinical patient-derived models for neuroblastoma, such as mice and zebrafish xenografts and spheroids- and organoids-based 3D in vitro tumor models. In addition, the authors discussed the the ethical considerations and technical limitations during engineering patient-derived neuroblastoma models and their potential application for dissecting tumor biology of neuroblastoma and preclinically screening candidate therapies. Overall, this manuscript would justify a publication in Journal of Personalized Medicine after the following concerns are addressed.

  1. The authors may add one or two Figures to better demonstrate the current processes and future considerations for personalized preclinical models.
  1. The numbering of different sections is not correct and there is no section 3.  In addition, Section title “2. Discussion” can be deleted.  Section 2.1 and 2.2 can be combined as the last section discussing the ethical considerations and technical limitations for creating patient-derived neuroblastoma models.
  1. The discussion on in vitro preclinical models for neuroblastoma can be expanded.

For example, the applications of microfluidic Organ-on-a-Chip for modeling neuroblastoma may be an alternative strategy.

Organ-on-a-Chip: A New Paradigm for Drug Development, Trends in Pharmacological Sciences; https://doi.org/10.1016/j.tips.2020.11.009.

Such potential of this new method has been demonstrated in several recent studies of modeling patient-relevant glioblastoma microenvironment for chemotherapy and immunotherapy screening, which may be used for modeling patient-specific neuroblastoma microenvironment.

Ex vivo profiling of pd-1 blockade using organotypic tumor spheroids. Cancer Discov. 2018, 8(2):196-215. doi:10.1158/2159-8290.CD-17-0833

Dissecting the immunosuppressive tumor microenvironments in Glioblastoma-on-a-Chip for optimized PD-1 immunotherapy. eLife 2020;9:e52253 DOI: 10.7554/eLife.52253

  1. Language and grammar should be carefully revised.

For example, the expansion of GD2 should be given at its first appearance.  Line 369 “CSF), anti-GD2 antibody with the differentiation agent cis-retinoic acid. Although this combination”; Line 374 “of activated Natural Killer (NK) cells could improve the response to anti-disialoganglioside (GD2)”.

In addition, “Line, 404 “Zebrafish modelling been undertaken successfully for”.

Author Response

Dear Sir,

Many thanks for your review of our manuscript.  Here follows a point-by-point response to your comments:

  1. The authors may add one or two figures
    • Following your suggestion, we have prepared a schematic illustration with a referenced legend to summarise the current process and considerations for personalised preclinical models.
  2. The numbering of different sections is not correct
    • Thank you for bringing this to our attention, and this mistake has been corrected.
  3. The discussion on in vitro models can be expanded
    • Thank you for providing the additional references.  They have been added to the review.
  4. Language and grammar should be carefully revised
    • Many thanks for noting these mistakes, they have now been corrected.

Yours sincerely

Dr Elizabeth Tucker